# Differences in aortic valve area measured on cardiac CT and echocardiography in patients with aortic stenosis

**Jooae Choe**[1], **Hyun Jung Koo**[1]*, **Se Jin Choi**[1], **Seung-Ah Lee**[2], **Dae-Hee Kim**[2], **Jong-Min Song**[2], **Duk-Hyun Kang**[2], **Jae-Kwan Song**[2], **Joon-Won Kang**[1], **Dong Hyun Yang**[1]

**1** Department of Radiology and Research Institute of Radiology, University of Ulsan College of Medicine, Asan Medical Center, Seoul, South Korea, **2** Division of Cardiology, Cardiac Imaging Center, University of Ulsan College of Medicine, Asan Medical Center, Seoul, South Korea

* elfin19@gmail.com

**Data Availability Statement:** The datasets generated during and/or analysed during the current study are not publicly available due to the policy of Institutional Ethics Committee but are

## Abstract

### Background

A certain proportion of patients with severe aortic stenosis (AS) present with discordant grading between different diagnostic modalities, which raises uncertainty about the true severity of AS. The aim of this study was to compare the aortic valve area (AVA) measured on CT and echocardiography and demonstrate the factors affecting AVA discrepancies.

### Methods

Between June 2011 and March 2016, 535 consecutive patients (66.83±8.80 years, 297 men) with AS who underwent pre-operative cardiac CT and echocardiography for aortic valve replacement were retrospectively included. AVA was obtained by AVA on echocardiography ($AVA_{echo}$) and CT ($AVA_{CT}$) using a measurement of the left ventricular outflow tract on each modality and correlations between those measures were evaluated. Logistic regression analysis was performed to identify factors affecting the discordance for grading severe AS.

### Results

The $AVA_{CT}$ and $AVA_{echo}$ showed a high correlation (r: 0.79, $P$ <0.001) but $AVA_{CT}$ was larger than the $AVA_{echo}$ (difference 0.26 cm$^2$, $P$ <0.001). By using the cut-off values of $AVA_{CT}$ (<1.2 cm$^2$) and $AVA_{echo}$ (<1.0 cm$^2$) for diagnosing severe AS, the BSA (odds ratio [OR]: 68.03, 95% confidence interval [CI]: 5.45–849.99; $P$ = 0.001), $AVA_{echo}$ (OR: 1.19, 95%CI: 1.14–1.24; $P$ <0.001), tricuspid valve morphology (OR: 2.83, 95%CI: 1.23–6.50; $P$ = 0.01), and normalized annulus area (OR: 1.02; 95%CI:1.02–1.03; $P$ <0.001) were significant factors associated with the discordance between the $AVA_{echo}$ and $AVA_{CT}$.

### Conclusion

Patients with larger BSA, $AVA_{echo}$, and annulus, and tricuspid valve morphology were associated with the AVA discordance between the echocardiography and CT.

available from the corresponding author (radkoo@amc.seoul.kr) on reasonable request, or from the following point of contact: - Name: Hong Min Oh - Affiliation: Asan Medical Center, Medical Imaging and Intelligent Reality Lab, Data management part - E-mail: ohm0124@gmail.com.

**Funding:** This work was supported by the National Research Foundation of Korea (NRF) grant funded by the Korea government (MSIT) (No.2022R1A5A1022977).

**Competing interests:** The authors have declared that no competing interests exist.

**Abbreviations:** AS, aortic stenosis; AUC, areas under the receiver operating characteristic curve; AVA, aortic valve area; AVR, aortic valve replacement; BSA, body surface area; BNP, B-type natriuretic peptide; CT, computed tomography; DLP, dose length product; ED, effective dose; ICC, intraclass correlation; IQR, interquartile range; LF/ LG, low-flow/low-gradient; LVOT, left ventricular outflow tract; PG, pressure gradient; TAVR, transcatheter aortic valve replacement; TTE, transthoracic echocardiography; VTI, velocity time integral.

Complementary use of CT with echocardiography for grading severe AS could be helpful in such conditions.

## 1. Introduction

Aortic stenosis (AS) is the most common valvular disease, and severe or symptomatic AS needs surgical or transcatheter aortic valve replacement (TAVR) [1, 2]. Unlike surgical aortic valve replacement (AVR), intraprocedural valve size evaluation is not possible in TAVR, and measuring the appropriate aortic annulus size is critical for the patient's outcome. To avoid complications in TAVR, such as aortic annulus rupture, paravalvular leak, and coronary artery obstruction, careful evaluation of the ventriculo-aortic transition is crucial. Moreover, in patients suspected of having significant AS, assessing the degree of AS is important in making decisions on stratified treatments and the timing for surgical intervention since the risk of waiting increases in correlation to the degree of AS [3]. A recent study demonstrated that surgical intervention could improve survival even among asymptomatic patients with severe AS [4]. The decision to perform surgery in an asymptomatic patient requires careful weighting of the risks of early AVR against those of observation.

The evaluation of the aortic apparatus and severity of AS is usually assessed by a transthoracic echocardiography (TTE) [5]. In a Doppler echocardiography, the single diameter of the aortic annulus (left ventricular outflow tract [LVOT] cross-sectional area) was used to calculate the aortic valve area (AVA). However, the aortic apparatus is a complex structure, and fibrocalcific changes of the aortic valve (AV) make evaluating the AV and annulus size on a 2D echocardiography alone difficult [6]. Due to the need for accurate preprocedural measurement of the aortic annular size, 3D-echocardiography and multidetector cardiac computed tomography (CT) have been widely used as potential alternatives to improve annulus measurements. On cardiac CT scans, we can obtain an AVA not only by the continuity equation ($AVA_{CT}$) but also by the planimetry method drawing the inner margin of the aortic cusps ($AVA_{plani}$). Several studies showed a good correlation between the CT-derived AVA and $AVA_{echo}$ calculated with the continuity equation using TTE measurements [7–10].

However, the AVA obtained by CT did not improve the correlation with the transaortic pressure gradient and yielded a greater AVA value than the $AVA_{echo}$; thus, a larger $AVA_{CT}$ cut-off value was recommended for diagnosing severe AS [11–13]. Previous studies suggested several explanations for the differences between the $AVA_{CT}$ and $AVA_{echo}$. One study suggested that a smaller $AVA_{echo}$ is related to the underestimation of the LVOT area with a single antero-posterior diameter of the LVOT, where the LVOT is an elliptical rather than round in aortic valvular disease [14]. Another study's proposed explanation was that the difference might show the difference in the maximum anatomical AVA and functional vena contracta, the narrowest portion of the stenotic flow that reflects a pressure gradient [15]. A third explanation is that on a CT scan, the AVA is measured at the systolic phase, whereas the average systolic phase parameters were used in the continuity equation [16].

It is unknown which modality, echocardiography or CT, is more accurately in grading AS, but the downgrade in diagnosis from severe AS by echocardiography to moderate AS by CT may change the timing of surgical treatments and could even reduce the number of patients that require AVR or TAVR. In addition, the effect of fibrocalcific changes of the AV and other echocardiographic characteristics in the discordance of classifying AS based on the $AVA_{echo}$ and $AVA_{CT}$ has not been sufficiently evaluated. Thus, the aims of our study are to confirm the correlation between the $AVA_{echo}$ and CT-derived AVA and to evaluate the factors resulting in the discordance between $AVA_{echo}$ and $AVA_{CT}$ when classifying severe AS.

## 2. Methods

### 2.1. Study patients

This retrospective study was approved by the Institutional review board (IRB) of our hospital (approval number: 2018–0233) and informed consent was waived. Between June 2011 and March 2016, 781 patients underwent surgical AVR. Exclusion criteria included patients with concomitant significant aortic regurgitation (qualitatively defined as moderate or higher degree based on echocardiographic findings; n = 177), quadricuspid AV (n = 1), or had no available preoperative cardiac CT scan (n = 47) or multiphase CT scan data (n = 21). Finally, 535 patients were included in the study. Clinical characteristics such as age, body surface area (BSA), hypertension, atrial fibrillation, heart failure, B-type natriuretic peptide (BNP), and outcomes, echocardiography parameters, and preoperative cardiac CT data were collected.

### 2.2. Echocardiography

All patients underwent a transthoracic echocardiography preoperatively using commercially available ultrasound machines with 3–5 MHz real-time transducers (iE33, EPIC; Philips Medical Systems, Andover, MA; Vivid 7, E9, General Electric Healthcare, Waukesha, WI, USA). Expert cardiologists obtained conventional two-dimensional and Doppler images according to the American Society of Echocardiography recommendations [17]. Color Doppler or three-dimensional images was obtained when clinically necessary. AS severity was semi-quantitatively measured and graded using four grades. Left ventricular ejection fraction (LVEF), AV velocity time integral (VTI), LVOT VTI, AVA, and volumetric parameters, including end-systolic volume and end-diastolic volume, were calculated. The maximal and mean pressure gradient (PG) across the AV were estimated using a modified Bernoulli equation, and the AVA was calculated from the continuity equation with VTI ($AVA_{echo}$; Fig 1). Classic low-flow/low-gradient (LF/LG) severe AS was defined as $AVA_{echo}$ less than 1 cm$^2$ but with a low gradient (<40 mmHg). Low-gradient severe AS with preserved LVEF was defined as paradoxical LF/LG AS.

### 2.3. Cardiac CT protocol and image analysis

A cardiac CT scan was performed preoperative using a second-generation dual-source CT scanner (Somatom Definition Flash; Siemens Medical Solutions, Forchheim, Germany).

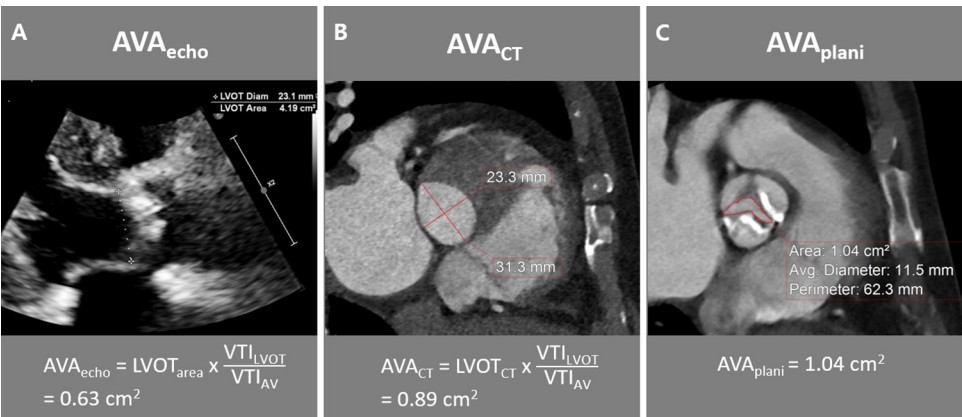

**Fig 1. An example of aortic valve area (AVA) measurement using echocardiography or CT in a 74-year-old male who diagnosed as severe stenosis of fused bicuspid aortic valve.** AVA, aortic valve area; LVOT, left ventricular outflow tract; VTI, velocity time integral.

Unless contraindicated, 2.5 mg oral bisoprolol (Concor; Merck, Darmstadt, Germany) was administered to patients with heart rates greater than 75 beats/min one hour prior to the CT scan for coronary artery evaluation. A bolus of 60–80 mL of nonionic, iodinated contrast material (Iomeron; Bracco Imaging SpA, Milan, Italy) was injected using a power injector (Stellant D; Medrad, Indianola, PA, USA) at 3.0 mL/s, followed by 40 mL of a 30:70 mixture of contrast and saline using the bolus tracking method (ascending aorta; trigger threshold level 100 HU; scan delay, 8 s). In our center, for baseline work up in patients with aortic valvular disease, multiphase cardiac CT is routinely performed prior to making the decision between surgery or TAVR as multiphase cardiac CT can provide information regarding the exact motion of aortic leaflets as well as morphological information regarding the aortic root and CT findings are also important in determining those treatment options. Retrospective electro-cardiogram-gated scanning was performed with a 20% tube current modulation (dose pulsing windows, 20%–70% of the R-R interval) to minimize the radiation dose. In patients with arrhythmia, 0%–90% of the R-R interval was used. The tube voltage and the tube current-time product were adjusted for body size. The scan parameters were as follows: tube voltage, 80–120 kV; tube current, 160–360 mAs; pitch, 0.17–0.38; detector collimation, 64 × 0.6 mm and gantry rotation time, 280 ms. For preoperative evaluation of the aorta, most patients underwent CT of the thorax and abdomen and cardiac CT. Therefore, the dose length product (DLP) and the effective dose (ED) values were slightly higher compared to those used for routine cardiac CT scans. In all patients, the mean ± standard deviation DLP for the cardiac CT scans was 1187.2 ± 547.7 mGy·cm, and the mean ED was 16.6 ± 7.7 mSv.

Post-processing was conducted using an external workstation (AquariusNet; TeraRecon, Foster City, CA, USA) using multiphase CT data sets reconstructed by a 10% R-R interval. CT characteristics such as AV morphology (tricuspid, bicuspid with raphe, bicuspid without raphe), $AVA_{CT}$, $AVA_{plani}$, aortic annulus diameter, circularity (minimum diameter/maximum diameter), perimeter, and area, and diameters of the sinus of Valsalva, sinotubular junction, and ascending aorta tubular portion were measured by two radiologists in consensus. $AVA_{CT}$ was calculated by using the LVOT area measured on CT (3D circular LVOT area approximated by $\pi r^2$ with average diameter used for $r$) in the continuity equation with VTI at LVOT and transaortic flow on echocardiography (a hybrid of measures from both CT and echocardiography; $AVA_{CT} = LVOT_{CT} \times VTI_{LVOT}/VTI_{Ao}$; Fig 1). For $AVA_{plani}$, the following steps were performed: 1) selecting the end-systolic phase showing maximal aortic opening; 2) generating an AV in the en-face view from the sinuotubular junction to LVOT level using 1 mm thick images and 3) measuring the largest AVA on an image with 5–10 mm slab thickness, generated from a multiplanar reformation to optimally visualize the lines of the AV tips. To evaluate the reliability of the CT measurements, a third experienced radiologist measured the CT parameters in 100 randomly selected cases, and an interobserver agreement was determined. Observers were blinded to the clinical data, including echocardiography findings and operation records.

## 2.4. Statistical analysis

Continuous variables were expressed as mean ± standard deviation or as median and inter-quartile range (IQR). Categorical variables are presented as numbers and percentages. Measurement variability was assessed by using intraclass correlation (ICC) for agreement between the image readers. The AVA measured on the echocardiography ($AVA_{echo}$) and CT ($AVA_{CT}$) were compared using a paired Student $t$- test. Agreement between the two modalities were assessed using ICC and the mean absolute difference was plotted using the Bland-Altman analysis to assess measurement differences in the $AVA_{echo}$ and $AVA_{CT}$. Correlation between the $AVA_{echo}$ and $AVA_{CT}$ was analyzed using the Pearson's correlation coefficient (r).

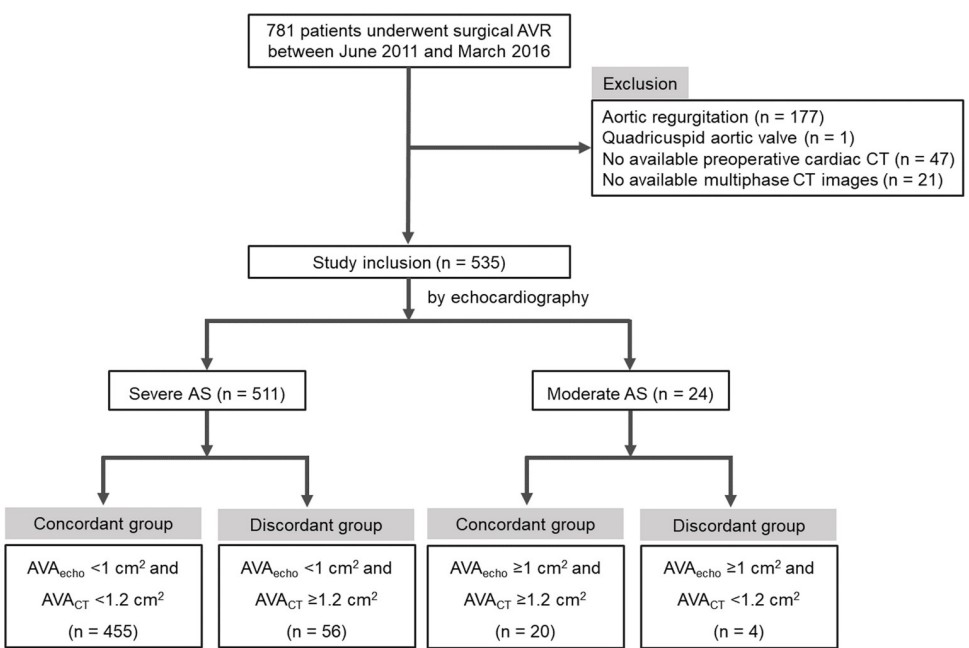

**Fig 2. Flow chart of patient inclusion.** AS, aortic stenosis; AVR, aortic valve replacement; AVA, aortic valve area.

Patients were classified into either the concordant group ($AVA_{echo} < 1.0$ cm$^2$ and $AVA_{CT} < 1.2$ cm$^2$) or the discordant group ($AVA_{echo} < 1.0$ cm$^2$ and $AVA_{CT} \geq 1.2$ cm$^2$) for grading severe AS [11–13]. The two groups were compared using the Student's *t*-test and Chi-square test or Fisher's exact test. Logistic regression analysis was performed to identify clinical and imaging parameters associated with the discordance group. Areas under the receiver operating characteristic curve (AUCs) were generated for each significant variable from a multivariable analysis. The cut-off values of the various parameters were identified and tested to maximize the Youden index (sensitivity + specificity − 1). All statistical tests were two-sided, and a *P* value less than 0.05 was used as a threshold for identifying a significant difference. Statistical analysis was performed using commercial software (SPSS, version 23; SPSS, Chicago, IL, USA).

## 3. Results

### 3.1. Patient characteristics

A total of 535 patients (mean age, 66.83 ± 8.80 years of age, 297 men) with AS who underwent preoperative cardiac CT and echocardiography for AVR were included in this study. Among the total patients, 95.51% (511/535) were diagnosed as severe AS, and 4.49% (24/535) were diagnosed as moderate AS on echocardiography (Fig 2). The median follow-up period was 4.13 years, and major adverse cardiac and cerebrovascular events (MACCE) occurred in 44 (8.22%) patients (Table 1).

### 3.2. Discordance between $AVA_{echo}$ and $AVA_{CT}$ in classifying severe AS

The inter-reader agreement of the CT measurements ranged from an ICC of 89.20–98.60 ($P < 0.001$). The correlation between $AVA_{echo}$ and $AVA_{CT}$ showed high positive correlation ($r = 0.79$, $P < 0.001$), whereas the correlation between $AVA_{plani}$ and $AVA_{echo}$ was moderate ($r = 0.52$, $P < 0.001$). The $AVA_{CT}$ was larger than the $AVA_{echo}$ (0.91 ± 0.30 vs. 0.65 ± 0.18 cm$^2$,

**Table 1. Clinical characteristics and echocardiographic findings of study patients.**

| Characteristics | Numbers |
|---|---|
| Age, years[†] | 66.83 ± 8.80 |
| Male | 297 (55.51) |
| BSA, m$^2$[*] | 1.65 ± 0.17 |
| Hypertension | 291 (54.39) |
| DM | 132 (24.67) |
| Atrial fibrillation | 79 (14.77) |
| Heart failure | 40 (7.48) |
| Aortic aneurysm | 97 (18.13) |
| PCI or CABG | 126 (23.55) |
| Concomitant aortic replacement | 77 (14.39) |
| B-type natriuretic peptide, pg/mL[*] | 343.71 ± 726.24 |
| Blood urea nitrogen, mg/dL[*] | 18.37 ± 8.20 |
| Creatinine, mg/dL[*] | 1.02 ± 0.96 |
| Echocardiography | |
| LVEF, %[*] | 59.51 ± 10.76 |
| Peak velocity, m/s[*] | 4.88 ± 0.91 |
| Mean PG, mmHg[*] | 60.03 ± 22.85 |
| LVMI, g/m$^2$[*] | 134.44 ± 35.61 |
| Aortic valve VTI, cm[*] | 120.39 ± 28.21 |
| LVOT VTI, cm[*] | 21.28 ± 4.21 |
| LVOT diameter, mm[*] | 21.09 ± 1.62 |
| AVA$_{echo}$, mm$^2$[*] | 64.71 ± 18.30 |
| ESVI, mL/m$^2$[*] | 28.87 ± 18.20 |
| EDVI, mL/m$^2$[*] | 67.72 ± 24.69 |
| Systemic arterial compliance, mL/m$^2$/mmHg[*] | 0.80 ± 0.34 |
| Valvulo-arterial impedance (Zva), mmHg/mL/m$^2$[*] | 5.21 ± 1.64 |
| Degree of AS | |
| Severe AS | 511 (95.51) |
| High gradient severe AS | 438 (81.87) |
| Classic LF-LG AS | 18 (3.36) |
| Paradoxical LF-LG AS | 55 (10.28) |
| Moderate AS | 24 (4.49) |
| CT findings | |
| Valve morphology | |
| Tricuspid | 260 (48.60) |
| Bicuspid with raphe | 131 (24.39) |
| Bicuspid without raphe | 144 (26.92) |
| AVA calcium score[*] | 2894.91 ± 1845.45 |
| AVA$_{plani}$, mm$^2$[*] | 90.32 ± 25. 36 |
| Calculated AVA$_{CT}$[*] | 90.89 ± 29.51 |
| Aortic annulus | |
| Circularity, %[*] | 81.45 ± 7.43 |
| Mean diameter, mm[*] | 25.01 ± 2.66 |
| Perimeter, mm[*] | 79.90 ± 8.44 |
| Area, mm$^2$[*] | 486.43 ± 102.89 |
| Sinus of Valsalva diameter, mm[*] | 36.66 ± 4.57 |
| Sinotubular junction diameter, mm[*] | 31.07 ± 4.74 |

(*Continued*)

**Table 1.** (Continued)

| Characteristics | Numbers |
|---|---|
| Ascending aorta tubular portion, mm* | 40.65 ± 6.51 |
| Surgical valve size, mm* | 22.17 ± 2.12 |
| Surgical valve type | |
| Bioprosthetic valve | 102 (19.07) |
| Mechanical valve | 433 (80.93) |
| Follow-up duration, year* | 4.13 ± 1.73 |
| Operative mortality (30-day mortality) | 8 (1.50) |
| MACCE including cardiovascular death | 44 (8.22) |
| Overall mortality | 74 (13.83) |

Note.–Data are numbers and the percentages are in parentheses.

*Mean ± standard deviation. AS, aortic stenosis; AVA, aortic valve area; BSA, body surface area; CABG, coronary artery bypass grafting; DM, diabetes mellitus; EDVI, end-diastolic volume index; ESVI, end-systolic volume index; LF/LG, low-flow and low-gradient; LVEF, left ventricular ejection fraction; LVMI, left ventricular mass index; LVOT, left ventricular outflow tract; MACCE, major adverse cardiac and cerebrovascular event; PCI, Percutaneous coronary intervention; PG, pressure gradient; VTI, velocity time integral.

$P < 0.001$) and the mean difference between $AVA_{CT}$ and $AVA_{echo}$ was 0.26 cm$^2$ (95% CI, 0.25 −0.27 cm$^2$, $P < 0.001$) (Fig 3). The difference between the $AVA_{CT}$ and $AVA_{echo}$ also increased as the $AVA_{echo}$ increased ($P < 0.001$).

Among the patients who were diagnosed as having severe AS by an echocardiography, 10.96% (56/511) of these patients had discordance between the $AVA_{echo}$ and $AVA_{CT}$ (Table 2).

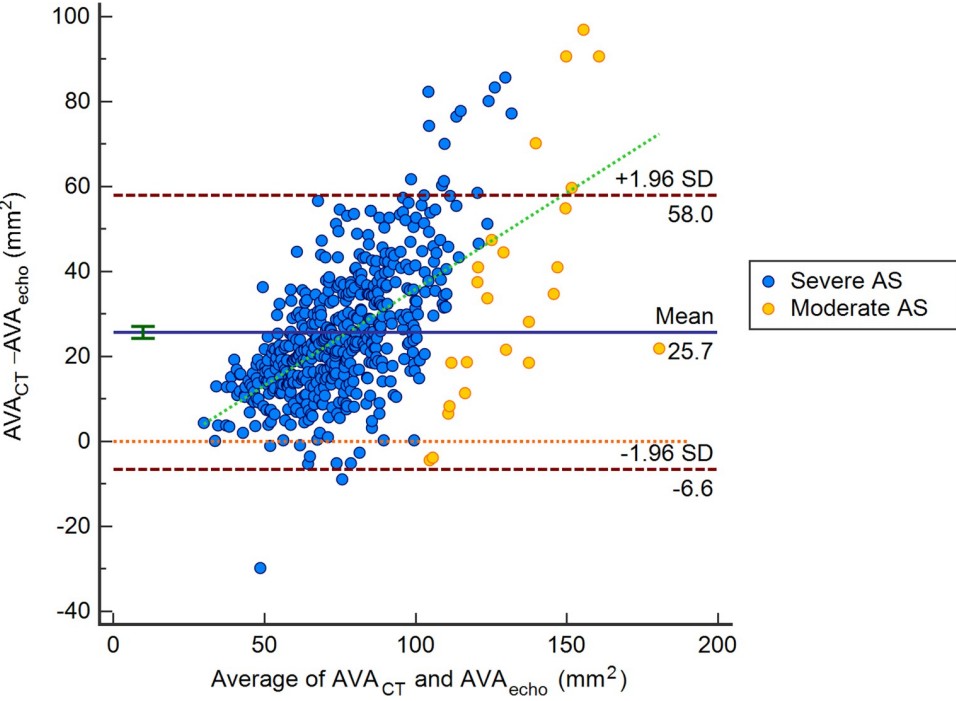

**Fig 3. Bland-Altman plot of $AVA_{echo}$ and $AVA_{CT}$.** AS, aortic stenosis; AVA, aortic valve area; SD, standard deviation.

**Table 2. Clinical and echocardiographic characteristics of concordant and discordant groups between AVA values measured by echocardiography and CT in severe AS.**

| Characteristics | Concordant group (n = 455) | Discordant group (n = 56) | P value |
|---|---|---|---|
| | $AVA_{echo}$ <1.0 cm² and $AVA_{CT}$ <1.2 cm² | $AVA_{echo}$ <1.0 cm² and $AVA_{CT}$ ≥1.2 cm² | |
| Age, years[*] | 66.84 ± 8.80 | 67.11 ± 8.61 | 0.83 |
| Male | 236 (51.87) | 44 (78.57) | < 0.001 |
| BSA, m²[*] | 1.63 ± 0.16 | 1.72 ± 0.16 | 0.001 |
| Hypertension | 240 (52.75) | 33 (58.93) | 0.38 |
| Atrial fibrillation | 65 (14.29) | 8 (14.29) | > 0.99 |
| PCI or CABG | 106 (23.30) | 11 (19.64) | 0.85 |
| Rheumatic valvular disease | 46 (10.11) | 2 (3.57) | 0.15 |
| B-type natriuretic peptide, pg/mL[†] | 106.00 (48.00–287.50) | 63.00 (31.00–254.00) | 0.13 |
| Echocardiography | | | |
| LVEF, %[*] | 59.83 ± 10.54 | 58.71 ± 11.54 | 0.46 |
| Peak velocity, m/s[*] | 4.98 ± 0.89 | 4.63 ± 0.74 | 0.004 |
| Peak PG, mmHg[*] | 101.82 ± 35.89 | 87.54 ± 56.82 | 0.004 |
| Mean PG, mmHg[*] | 62.12 ± 22.97 | 52.30 ± 17.60 | 0.002 |
| LVMI, g/ m²[*] | 135.05 ± 36.0 | 132.38 ± 35.31 | 0.60 |
| AV VTI, cm[*] | 125.24 ± 26.63 | 97.58 ± 19.14 | < 0.001 |
| LVOT VTI, cm[*] | 21.09 ± 4.17 | 22.01 ± 4.05 | 0.12 |
| LVOT diameter, mm[*] | 21.02 ± 1.52 | 21.29 ± 1.51 | 0.20 |
| LVOT diameter/BSA, mm[*] | 12.96 ± 1.33 | 12.47 ± 1.11 | 0.008 |
| $AVA_{echo}$, mm²[*] | 60.02 ± 13.70 | 81.68 ± 8.77 | < 0.001 |
| ESVI, mL/m²[*] | 28.37 ± 17.91 | 30.68 ± 19.07 | 0.37 |
| EDVI, mL/m²[*] | 67.21 ± 23.35 | 70.49 ± 26.52 | 0.08 |
| SAC, mL/m²/mmHg[*] | 0.81 ± 0.34 | 0.76 ± 0.28 | 0.26 |
| Valvulo-arterial impedance (Zva), mmHg/mL/m²[*] | 5.24 ± 1.64 | 5.06 ± 1.66 | 0.43 |
| Subgroups | | | 0.04 |
| Severe AS | 395 (86.81) | 43 (76.79) | 0.04[‡] |
| Classic LF-LG AS | 13 (2.86) | 5 (8.93) | |
| Paradoxical LF-LG AS | 47 (10.33) | 8 (14.29) | |
| Surgical valve size, mm[*] | 21.99 ± 2.09 | 23.32 ± 1.76 | < 0.001 |
| Postoperative findings | | | |
| LVEF, %[*] | 59.64 ± 9.35 | 56.82± 9.41 | 0.04 |
| Peak velocity, m/s[*] | 2.65 ± 0.58 | 2.40 ± 0.66 | 0.004 |
| Peak PG, mmHg[*] | 28.41 ± 10.92 | 24.15 ± 11.14 | 0.008 |
| Mean PG, mmHg[*] | 15.77 ± 6.70 | 13.46 ± 6.39 | 0.02 |
| LVMI, g/m²[*] | 114.41 ± 31.17 | 118.18 ± 35.92 | 0.41 |
| MACCE | 36 (7.91) | 7 (12.50) | 0.24 |
| Overall mortality | 63 (13.85) | 8 (14.30) | 0.93 |

Note.–Data are numbers and the percentages in parentheses.

[*]Data are mean ± standard deviation.

[†]Data are median and the interquartile range in parentheses.

[‡]Comparison between patients with severe AS and LF/LG AS (classic or paradoxical). AS, aortic stenosis; AVA, aortic valve area; BSA, body surface area; CABG, coronary artery bypass graft; EDVI, end-diastolic volume index; ESVI, end-systolic volume index; LF/LG, low-flow and low-gradient; LVEF, left ventricular ejection fraction; LVMI, left ventricular mass index; LVOT, left ventricular outflow tract; MACCE, major adverse cardiac and cerebrovascular event; PCI, percutaneous coronary intervention; PG, pressure gradient; SAC, systemic arterial compliance; VTI, velocity time integral.

Male patients were more common than female patients in the discordant group (78.57% vs. 51.87%, $P < 0.001$). The BSA was larger in the discordant group than in the concordant group ($1.72 \pm 0.16$ m$^2$ vs. $1.63 \pm 0.16$ m$^2$, $P = 0.001$). Bicuspid AV was less frequent in the discordant group than in the concordant group (37.50% vs. 53.63%; $P < 0.02$). The mean AVA$_{echo}$ was larger in the discordant group than in the concordant group ($81.68 \pm 8.77$ mm$^2$ vs. $60.02 \pm 13.70$ mm$^2$, $P < 0.001$). LF/LG AS was more frequent in the discordant group compared to the concordant group (23.21% vs. 13.19%, $P = 0.04$; Table 2 and S1 Table in S1 File). Peak velocity, peak pressure gradient (PG) and mean PG was lower in the discordant group than in the concordant group (for all, $P < 0.05$). The CT measurements of the calculated AVA$_{CT}$, AVA$_{plani}$, annulus diameter, area, and perimeter, mean diameters of the LVOT, sinus of Valsalva, and ST junction, all were significantly larger in the discordant group than in the concordant group, both before normalization and after normalization to the BSA ($P < 0.05$) (Table 3). In the discordant group, the tricuspid AV was the most common valve morphology and was more frequent than the concordance group (62.50% vs. 45.38%, $P = 0.01$). There was no significant difference in the AVA calcium score and aortic annulus circularity between the discordant and concordant groups ($P = 0.180$ and $P = 0.21$, respectively).

Among the patients who were diagnosed as moderate AS on echocardiography, 16.67% (4/24) of these patients showed discordance between the AVA$_{echo}$ and AVA$_{CT}$ (AVA$_{echo}$ $\geq 1.0$ cm$^2$ and AVA$_{CT}$ $< 1.2$ cm$^2$). The median AVA$_{CT}$ was 151.45 mm$^2$ (IQR: 140.61–179.24) for the concordant group and 108.88 mm$^2$ (IQR: 103.07–114.68) for the discordant group, which was significantly smaller in the discordant group ($P = 0.002$). The discordant group showed higher LVEF and AV VTI, lower end-systolic volume index, and smaller perimeter and area of the aortic annulus (for all, $P < 0.05$) (S2 Table in S1 File).

### 3.3. Factors associated with the discordance between AVA$_{echo}$ and AVA$_{CT}$

The univariable logistic regression analysis indicated that male sex, BSA, peak velocity, peak PG, mean PG, AV VTI, normalized LVOT, AVA$_{echo}$, LF/LG AS, tricuspid AV, normalized AVA$_{CT}$ both normalized and unnormalized AVA$_{plani}$, aortic annulus measurements (diameter, perimeter, annulus area) and diameters of sinus of Valsalva and ST junction were significant factors associated with the discordance between the AVA$_{echo}$ and AVA$_{CT}$ ($P < .05$) in patients with severe AS (AVA$_{echo}$ $< 1.0$ cm$^2$) (Table 4). In the multivariable analysis, BSA (odds ratio [OR]: 68.03, 95% confidence interval [CI]:5.45–849.99; $P = 0.001$), AVA$_{echo}$ (OR: 1.19, 95%CI:1.14–1.24; $P < .001$), tricuspid valve morphology (OR: 2.83, 95%CI:1.23–6.50; $P = 0.01$) and the annulus area normalized to the BSA (OR: 1.02, 95%CI:1.02–1.03; $P < 0.001$) were significant independent factors for the discordance between the AVA$_{echo}$ and AVA$_{CT}$ (Table 5). The optimal cut-offs for predicting discordance between the AVA$_{echo}$ and AVA$_{CT}$ were $> 1.74$ m$^2$ for the BSA (AUC: 0.65, sensitivity: 46.43%, specificity: 76.26%), $> 73.44$ mm$^2$ for the AVA$_{echo}$ (AUC: 0.91 sensitivity: 83.93%, specificity: 84.18%), and $> 305.50$ mm$^2$ for annulus area normalized to BSA (AUC: 0.72, sensitivity: 73.21%, specificity: 67.03%).

## 4. Discussion

Our study demonstrated that 10.96% (56/511) of patients with severe AS based on echocardiography showed discordant grading of severe AS between echocardiography and CT. Patients with larger BSA, AVA$_{echo}$, tricuspid valve morphology, and annulus area normalized to the BSA were significant factors that were associated with the discordant grading of AVA between echocardiography and CT. The combined use of CT and echocardiography for grading severe AS should be emphasized and might be helpful in these patients.

**Table 3. CT findings of concordant and discordant groups between AVA values measured by echocardiography and CT in severe AS.**

| CT findings | Concordant group (n = 455) | Discordant group (n = 56) | *P value* |
|---|---|---|---|
| | $AVA_{echo} <1.0$ cm$^2$ and $AVA_{CT}$ $<1.2$ cm$^2$ | $AVA_{echo} <1.0$ cm$^2$ and $AVA_{CT} \geq 1.2$ cm$^2$ | |
| Valve morphology | | | 0.011 |
| Tricuspid (%) | 211 (45.38) | 35 (62.50) | |
| Bicuspid with raphe (%) | 133 (28.60) | 6 (10.71) | |
| Bicuspid without raphe (%) | 111 (23.87) | 15 (26.79) | |
| LVOT mean diameter* | 24.62 ± 2.82 | 26.59 ± 2.69 | < 0.001 |
| AVA calcium score† | 2747.90 (1523.28–4153.45) | 2229.95 (1422.80–3734.95) | 0.18 |
| ln(AVA calcium score)† | 7.92 (7.33–8.33) | 7.71 (7.26–8.23) | 0.18 |
| $AVA_{plani}$, mm$^{2*}$ | 86.20 ± 22.53 | 109.16 ± 26.04 | < 0.001 |
| $AVA_{CT}$, mm$^{2*}$ | 82.06 ± 18.95 | 136.85± 26.71 | < 0.001 |
| Aortic annulus | | | |
| Circularity, %* | 81.66 ± 7.49 | 81.34 ± 7.95 | 0.21 |
| Maximal dimeter, mm* | 27.21 ± 3.06 | 30.42 ± 3.04 | < 0.001 |
| Mean diameter, mm* | 24.67 ± 2.52 | 27.38 ± 2.46 | < 0.001 |
| Perimeter, mm* | 78.92 ± 8.08 | 86.82 ± 7.97 | < 0.001 |
| Area, mm$^{2*}$ | 474.67 ± 96.84 | 570.70 ± 110.80 | < 0.001 |
| Sinus of Valsalva diameter, mm* | 36.34 ± 4.56 | 39.09 ± 4.01 | < 0.001 |
| Sinotubular junction diameter, mm* | 30.82 ±4.74 | 32.64 ± 4.36 | 0.007 |
| Ascending aorta tubular portion, mm* | 40.58 ± 6.31 | 40.83 ± 7.07 | 0.79 |
| Measurements normalized to BSA | | | |
| $AVA_{plani}$, mm$^{2*}$ | 52.97 ± 13.70 | 64.27 ± 16.97 | < 0.001 |
| $AVA_{CT}$, mm$^{2*}$ | 50.34 ± 11.27 | 80.03 ± 20.64 | < 0.001 |
| Aortic annulus | | | |
| Maximal dimeter, mm* | 16.74 ± 1.95 | 17.80 ± 1.92 | < 0.001 |
| Mean diameter, mm* | 15.18 ± 1.65 | 16.02 ± 1.53 | < 0.001 |
| Perimeter, mm* | 48.56 ± 5.22 | 50.80 ± 4.85 | 0.002 |
| Area, mm$^{2*}$ | 290.89 ± 54.29 | 332.28 ± 55.43 | < 0.001 |
| Sinus of Valsalva diameter, mm* | 22.37 ± 2.92 | 22.88 ± 2.50 | 0.21 |
| Sinotubular junction diameter, mm* | 18.97 ± 2.98 | 19.10 ± 2.65 | 0.76 |
| Ascending aorta tubular portion, mm* | 25.02 ± 4.29 | 23.93 ± 4.47 | 0.07 |

Note.–Data are numbers and the percentages in parentheses.

*Data are mean ± standard deviation.

†Data are median and the interquartile range in parentheses. AS, aortic stenosis; AVA, aortic valve area; BSA, body surface area; LVOT, left ventricular outflow tract.

As demonstrated by Clavel et al., the $AVA_{CT}$ was larger than the $AVA_{echo}$, and larger cut-point of 1.2 cm$^2$ should be used for grading severe AS on CT [13]. Although we applied the larger cut-point of 1.2 cm$^2$ for grading severe AS on CT compared to the cut-point of 1.0 cm$^2$ on echocardiography, 10.96% of patients showed discordance between grading of severe AS on the CT and echocardiography. Although it is unknown which modality can more accurately grade severe AS in the discordant group, changes from the diagnosis of severe AS on the echocardiography to moderate AS by CT may affecting the timing of AV surgery or intervention.

Inaccurate measurements of the LVOT diameters on echocardiography and low stroke volume can affecting the inconsistent grading of severe AS [18]. In a previous study, 52% of normal flow–low gradient and 12% of LF/LG severe AS are reclassified into moderate AS [18]. In

**Table 4. Univariate logistic regression analysis of clinical and CT characteristics to predict the discordance between the aortic valvular areas measured by echocardiography and CT.**

| Variables | OR (95% CI) | P value |
|---|---|---|
| Age, years | 1.00 (0.97–1.04) | 0.83 |
| Male | 3.40 (1.75–6.61) | < 0.001 |
| BSA, $m^2$ | 21.1 (3.36–132.6) | 0.001 |
| Echocardiography | | |
| Peak velocity, m/s | 0.64 (0.46–0.87) | 0.005 |
| Peak PG | 0.99 (0.97–1.00) | 0.004 |
| Mean PG | 0.98 (0.97–0.99) | 0.002 |
| AV VTI, cm | 0.95 (0.93–0.96) | < 0.001 |
| LVOT diameter/BSA, mm | 0.74 (0.59–0.93) | 0.009 |
| $AVA_{echo}$, $cm^2$ | 1.15 (1.11–1.19) | < 0.001 |
| LF/LG AS (reference: severe AS) | 1.99 (1.01–3.92) | 0.05 |
| CT findings | | |
| Tricuspid morphology (reference: bicuspid) | 1.93 (1.09–3.41) | 0.02 |
| LVOT mean diameter, mm | 1.27 (1.15–1.40) | < 0.001 |
| $AVA_{plani}$, $cm^2$ | 1.04 (1.03–1.05) | < 0.001 |
| Aortic annulus mean diameter, mm | 1.47 (1.31–1.64) | < 0.001 |
| Aortic annulus maximal diameter, mm | 1.34 (1.22–1.47) | < 0.001 |
| Aortic annulus perimeter, mm | 1.12 (1.08–1.16) | < 0.001 |
| Aortic annulus area, $mm^2$ | 1.01 (1.01–1.01) | < 0.001 |
| Sinus of Valsalva diameter | 1.14 (1.07–1.21) | < 0.001 |
| Sinotubular junction diameter, mm | 1.08 (1.02–1.13) | 0.008 |
| Normalized to BSA | | |
| $AVA_{plani}$/BSA, $mm^{2*}$ | 1.05 (1.03–1.07) | < 0.001 |
| $AVA_{CT}$/BSA, $mm^{2*}$ | 1.26 (1.19–1.34) | < 0.001 |
| Aortic annulus | | |
| Maximal dimeter/BSA, $mm^*$ | 1.29 (1.13–1.47) | < 0.001 |
| Mean diameter/BSA, $mm^*$ | 1.35 (1.14–1.59) | < 0.001 |
| Perimeter/BSA, $mm^*$ | 1.08 (1.03–1.14) | 0.003 |
| Area/BSA, $mm^{2*}$ | 1.01 (1.01–1.02) | < 0.001 |

AV, aortic valve; AVA, aortic valve area; BSA, body surface area; CI, confidence interval; LF/LG, low-flow and low-gradient; LVOT, left ventricular outflow tract; OR, odds ratio; PG, pressure gradient; VTI, velocity time integral.

our study, LF/LG AS was also more frequent in the discordant group than in the concordant group (23.21% vs. 13.19%; $P = 0.04$). Implementation of CT scans may be the important discriminatory method between true severe and moderate AS by minimizing the inaccurate

**Table 5. Multivariable logistic regression analysis of clinical and CT characteristics to predict the discordance between the aortic valvular areas measured by echocardiography and CT.**

| | OR (95%CI) | P value |
|---|---|---|
| BSA, $m^2$ | 68.03 (5.45–849.99) | 0.001 |
| $AVA_{ehco}$, $cm^2$ | 1.19 (1.14–1.24) | < 0.001 |
| Tricuspid morphology (reference: bicuspid) | 2.83 (1.23–6.50) | 0.014 |
| Annulus area normalized to BSA | 1.02 (1.02–1.03) | < 0.001 |

AVA, aortic valve area; BSA, body surface area; CI, confidence interval; OR, odds ratio.

measurements of AVA due to artifacts on the echocardiography from leaflet calcifications or anatomical assumptions of a circular shape of the LVOT in two-dimensional echocardiography along with stress echocardiography [19].

It has been suggested that the use of $AVA_{echo}$ index to the BSA for patients with either unusually small or large body sizes. However, indexing for body size is controversial primarily because the current algorithms for defining body size, such as the BSA, do not necessarily reflect the normal AVA in obese patients because the valve area does not increase with body size. Therefore, the AVA index can be underestimated in patients with a large BSA. In our study, large BSA and large EDVI suggested that large heart size was a significant factor in the discordance between $AVA_{echo}$ and $AVA_{CT}$ in grading severe AS. In these patients, measurement errors that may cause underestimation of the AVA, PG, and flow on the echocardiography could be prudentially re-evaluated with the combined use of CT [20].

In this study, the correlation between $AVA_{echo}$ and $AVA_{CT}$ showed a high positive correlation, whereas the correlation between $AVA_{plani}$ and $AVA_{echo}$ was moderate (r, 0.79 vs. 0.52). As a result of this finding, we further defined discordant and concordant groups based on AVACT by the continuity equation rather than $AVA_{plani}$. Accurately measuring AVA using a planimetry approach is limited due to the difficulty in drawing the maximal systolic opening area on 2D-thin slice thickness images as the tips of the aortic cusps are not contained to one 2D plane. On the other hand, using thick slice thickness images for planimetry area measurements makes it challenging to outline the margin of cusps due to the blurring of images which can lead to overestimating the valve opening area. Therefore, calculating AVA by incorporating velocity information from echocardiography and the size of LVOT on high spatial resolution cardiac CT images based on continuity equations can be more accurate to reflect the maximal systolic opening of AV.

Our study has several limitations. Firstly, the prognostic implications of the discordance between $AVA_{CT}$ and $AVA_{echo}$ could not be addressed in this study. This resulted from the discordant results between CT and echocardiography not being considered when determining the timing of AVR. Furthermore, as echocardiography was the standard exam, the effect of discrepancies between two modalities on the patients outcome could not be evaluated. Secondly, our cohort was limited to patients who underwent AVR and therefore does not evaluate the effect on mortality due to the natural course of disease according to AS severity. As our study included patients with only a narrow higher end (severe) of AS, as opposed to the lower end (mild) of AS, this can cause selection bias. Future studies focusing on assessing the modality of choice to establish the optimal time of surgery and the effect of AS severity grading using these different modalities would be of value in order to determine the clinical outcome of medical management compared to surgery. Second, we were not able to consider the effect of dynamic changes in the diameter of LVOT on measurement variability of $AVA_{echo}$, as the LVOT diameter measured during the mid-systolic phase was used for calculating the $AVA_{echo}$. Third, in this study, since the echocardiography or CT data were collected from the actual clinical records, AV morphology was described based on the previously described classification by Sievers et al [21]. However, the recently suggested new classification of AV morphology subclassified fused, 2-sinus, and partial fusion type of bicuspid AV [22]. However, little is known about whether the new classification is better or not to predict outcome in patients with AS. Future studies with the new classification of AVs would have a value.

## 5. Conclusion

Larger BSA, $AVA_{echo}$, tricuspid valve morphology, and annulus size were associated with discordance between grading severe AS based on AVA on echocardiography and CT. The

combined use of CT and echocardiography for grading AS should be emphasized and will be helpful in these patients.

## Supporting information

**S1 File.**
(DOCX)

## Author Contributions

**Conceptualization:** Hyun Jung Koo, Dong Hyun Yang.

**Data curation:** Jooae Choe, Hyun Jung Koo, Seung-Ah Lee.

**Formal analysis:** Jooae Choe, Hyun Jung Koo, Se Jin Choi.

**Investigation:** Jooae Choe, Hyun Jung Koo, Se Jin Choi.

**Methodology:** Jooae Choe, Hyun Jung Koo.

**Project administration:** Hyun Jung Koo.

**Resources:** Hyun Jung Koo.

**Software:** Jooae Choe, Hyun Jung Koo.

**Validation:** Hyun Jung Koo.

**Visualization:** Jooae Choe, Hyun Jung Koo.

**Writing – original draft:** Jooae Choe, Hyun Jung Koo, Se Jin Choi.

**Writing – review & editing:** Jooae Choe, Hyun Jung Koo, Seung-Ah Lee, Dae-Hee Kim, Jong-Min Song, Duk-Hyun Kang, Jae-Kwan Song, Joon-Won Kang, Dong Hyun Yang.

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
