## [Decision Letter · Decision Letter 0]

7 Oct 2022

PONE-D-22-24629Differences in Aortic Valve Area Measured on Cardiac CT and Echocardiography in Patients with Aortic StenosisPLOS ONE

Dear Dr. Koo,

Thank you for submitting your manuscript to PLOS ONE. After careful consideration, we feel that it has merit but does not fully meet PLOS ONE’s publication criteria as it currently stands. Therefore, we invite you to submit a revised version of the manuscript that addresses the points raised during the review process.

We look forward to receiving your revised manuscript.

Kind regards,

Tom Wang

Academic Editor

PLOS ONE

Journal Requirements:

Additional Editor Comments:

Thanks for your submission. In addition to reviewer comments below (or to repeat), please address the following questions/comments:

1. It appears the vast majority of patients at your hospital undergoing surgical AVR had pre-op cardiac CT with multiphase. Explain the rationale of this practice and specific benefits - most centers probably only do single phase pre-op cardiac CT, unless it is for TAVI workup. Also explain reasons why some patients didn’t get CT or multiphase CT (perhaps CKD)?

2. Were everybody with having AVR included in this study, or just those having isolated AVR. Was severe aortic stenosis an inclusion criteria? Was prior AVR, cardiac surgery, or endocarditis excluded? Explain the definition of concomitant significant AR (is this moderate or higher, or severe or higher, and how was this graded qualitative or quantitative)? Put in the limitations the selection bias that we are dealing with only a narrow higher end of aortic stenosis not the lower end.

3. Explain what method was used to calculate AVA on echo – was it continuity equation with VTI or peak velocity, how was stroke volume calculated (LVOT-VTI method or LVEDV-LVESV), or even planimetry. Continuity methods are expected to have discrepancy with anatomical methods for AVA, in addition to echo and CT having different measurements – please comment on these differences in the discussion.

4. You defined classic and paradoxical low flow low gradient AS in your methods along with severe AS and moderate AS – please present baseline/echo parameters in table 1 based on these subgroups also. Was there an association between the LFLG AS subgroups and AVA discordance discordance? What about calcium score (which can distort AVA-CT measurement) is it associated with the AVA discordance?

5. Add the following characteristics to table 1: diabetes, coronary heart disease, aortic aneurysm, NYHA class, echo: LVSVi, dimensionless index, LAVi, RVSP, RV function, CT: LVEDVi, LVESVi, LVSVi, LVEF; bioprosthetic or mechanical AVR, concomitant surgery, operative mortality (30-days).

6. In the Bland-Altman plot figure, please use average AVA echo and ct in the X axis.

7. 6. Another figure illustrating the AVAecho and AVA CT technique used would be appreciated for the journal’s medical audience.

8. Perform multivariable analysis of clinical, echo and CT parameters to predict overall mortality during follow-up (remember up to 7 covariates given 74 mortality events). Was CT valve parameters stronger predictor than echo?

9. Have a paragraph before limitations in the discussion called clinical implications. What are the main implications for your study? When and how do you recommend CT-AVA be used and what thresholds for severe? How does your study change current management?

10. Any items you can’t address above need to add to limitations.

Reviewers' comments:

Reviewer's Responses to Questions

**Comments to the Author**

1. Is the manuscript technically sound, and do the data support the conclusions?

Reviewer #1: Yes

Reviewer #2: Yes

2. Has the statistical analysis been performed appropriately and rigorously? 

Reviewer #1: I Don't Know

Reviewer #2: Yes

3. Have the authors made all data underlying the findings in their manuscript fully available?

Reviewer #1: No

Reviewer #2: Yes

4. Is the manuscript presented in an intelligible fashion and written in standard English?

Reviewer #1: No

Reviewer #2: Yes

5. Review Comments to the Author

Reviewer #1: 1-Retrospective study.

2- Data collected from medical records

3-Methods section lacks details on how the AoVA by CT was calculated. Did the authors used the CT maximal LVOT diameter, average area or planimetered LVOT area?.

4-Discordant valve area was defined when the difference between echo and CT was>0.2cm2. This occurred in a relatively small number of patients.

5-As in prior studies AV by CT was largen than echo AoVA.

6-The main objective of the study was to find the determinants form discrepant AoVA

7-In the bland Altman plot shown, the X axis shows the echo AoVA and not the average of Echo and CT as is tipically done.

8-Not clear what the unique contribution to this paper compared to similar ones that already exist in the literature.

Reviewer #2: The manuscript entitled "Differences in Aortic Valve Area Measured on Cardiac CT and Echocardiography in Patients with Aortic Stenosis" by Koo. et al. aimed to compare the aortic valve area AVA measured on CT and echocardiography, and demonstrate the factors that lead to discordant AVA readings between CT and echocardiography. From the 781 retrospectively selected patients, 95.51% were documented to have echocardiographic severe aortic stenosis and 10.96% of those had discordance between AVAecho and AVACT. Larger BSA, AVAecho, tricuspid AV morphology and indexed annulus area were associated with AVA discordance between echo and CT.

The authors have done a lot of work in collating this information and improving our knowledge in multimodality AS classification with few clarifications requested.

1. It should be highlighted in the text that the main comparison between AVAecho and AVACT utilizes continuity equation for both and thus are comparisons of an echo solely or a hybrid approach. This said, a hybrid approach measurement can also be simplified to the comparison of annular area in both modalities.

2. Furthermore, the discussion section sheds less light on AVAplani as a comparison and thus is confusing as the norm of a CT approach is usually to check planimetry area measurements. If that is not the case, it is then important to emphasize the differences within CT measurements whether measurement have been taken by the hybrid approach or by planimetry.

6. PLOS authors have the option to publish the peer review history of their article (what does this mean?). If published, this will include your full peer review and any attached files.

Reviewer #1: No

Reviewer #2: No

---

## [Author Response · Author response to Decision Letter 0]

19 Dec 2022

Point-by-point response to reviewers

Submission ID: PONE-D-22-24629

Manuscript title: "Differences in Aortic Valve Area Measured on Cardiac CT and Echocardiography in Patients with Aortic Stenosis"

In response to the reviewer’s comments, we have attempted to follow the editors and reviewer’s recommendations to our best ability. We have addressed several important matters that the reviewers raised, and we have revised the manuscript appropriately. We hope that our responses and revisions will alleviate the reviewers concerns and we would like to thank you for taking the time to review our manuscript.

Editor’s comments

Thanks for your submission. In addition to reviewer comments below (or to repeat), please address the following questions/comments:

E-1. It appears the vast majority of patients at your hospital undergoing surgical AVR had pre-op cardiac CT with multiphase. Explain the rationale of this practice and specific benefits - most centers probably only do single phase pre-op cardiac CT, unless it is for TAVI workup. Also explain reasons why some patients didn’t get CT or multiphase CT (perhaps CKD)?

Response: In our center, for baseline work up in patients with aortic valvular disease, multiphase CT is routinely used as a preoperative cardiac CT. Multiphase data can provide information regarding the exact motion of aortic leaflets as well as morphological information regarding the aortic root. Echocardiography may be sufficient in terms of determining AS severity, but as demonstrated in our study, in certain patients, echocardiography could be insufficient and prudentially re-evaluated with the combined use of CT. Although this cohort is a group of patients who underwent AVR, for patients with symptomatic AS that require intervention including surgical or procedural treatment, cardiac CT is routinely performed prior to making the decision between surgery or TAVI because CT findings are also important in determining those treatment options. Intraoperative valve sizing can be done by inserting a sizer directly in the aortic root, however, preoperative sizing on systolic phase CT data is useful for narrowing down as for a preliminary selection. Therefore, in our center, we usually perform multiphase cardiac CT with a 20% tube current modulation (dose pulsing windows, 20%–70% of the R-R interval) to minimize the radiation dose in patients with valvular heart disease. Those patients who were excluded from our cohort due to unavailable multiphase data were patients who underwent only single-phase cardiac CT for the purpose of evaluation of coronary artery disease, rather than valvular disease, or patients with missing multiphase data in our database storage. We provided additional description in the Materials and Methods for the rationale of performing multiphase cardiac CT in our center. 

Revised manuscript:

In our center, for baseline work up in patients with aortic valvular disease, multiphase cardiac CT is routinely performed prior to making the decision between surgery or TAVR as multiphase cardiac CT can provide information regarding the exact motion of aortic leaflets as well as morphological information regarding the aortic root and those CT findings are also important in determining those treatment options. Retrospective electrocardiogram-gated scanning was performed with a 20% tube current modulation (dose pulsing windows, 20%–70% of the R-R interval) to minimize the radiation dose. 

E-2. Were everybody with having AVR included in this study, or just those having isolated AVR. Was severe aortic stenosis an inclusion criteria? Was prior AVR, cardiac surgery, or endocarditis excluded? Explain the definition of concomitant significant AR (is this moderate or higher, or severe or higher, and how was this graded qualitative or quantitative)? Put in the limitations the selection bias that we are dealing with only a narrow higher end of aortic stenosis not the lower end.

Response: In this study, patients who underwent surgical AVR due to symptomatic moderate to severe AS were consecutively included. Among the total included patients, 95.51% (511/535) were diagnosed as severe AS, and 4.49% (24/535) were diagnosed as moderate AS based on echocardiography. 

Patients with AVR alone were searched from the electronic surgical records. Patients who previously underwent valvular surgeries were initially excluded. Then, patients with endocarditis (n = 11) or concomitant significant aortic regurgitation (AR) (n = 177) were excluded and significant AR was qualitatively defined as moderate or high degree based on echocardiographic findings. As patients usually underwent multiphase cardiac CT for preprocedural or presurgical evaluation, most patients who do not have AS or have a mild degree of AS do not undergo multiphase cardiac CT in clinical practice. Moreover, we also agree with your comment, that in our study cohort which included patients who underwent AVR, but only those with a narrow higher end of aortic stenosis (AS), not the lower end of AS. We included this point as a study limitation as it can lead to selection bias and the study results must be interpretated tentatively. 

Revised manuscript (Methods): Exclusion criteria included patients with concomitant significant aortic regurgitation (qualitatively defined as moderate or higher degree based on echocardiographic findings; n = 177), [ÿ]

Revised manuscript (Discussion): As our study included patients with only a narrow higher end of AS, as opposed to the lower end of AS, this can cause selection bias. Therefore, caution is needed when interpreting the study results.

E-3. Explain what method was used to calculate AVA on echo – was it continuity equation with VTI or peak velocity, how was stroke volume calculated (LVOT-VTI method or LVEDV-LVESV), or even planimetry. Continuity methods are expected to have discrepancy with anatomical methods for AVA, in addition to echo and CT having different measurements – please comment on these differences in the discussion.

Response: Thank you for your comment. We agree with what you have discussed. AVA on echocardiography was calculated using a continuity equation with VTI. We added an additional description for the AVAecho and AVACT in our revised manuscript. AVACT was also calculated by using a hybrid method incorporating information from both echocardiography and CT, with the LVOT area measured on CT in the continuity equation with VTI at LVOT and transaortic flow on echocardiography (AVACT = LVOTCT × VTILVOT/VTIAo). For the LVOT area, a circular LVOT area was estimated by πr2 using an average diameter. 

Revised manuscript (Methods): 

The maximal and mean pressure gradients (PG) across the AV were estimated using a modified Bernoulli equation, and the AVA was calculated from the continuity equation with VTI (AVAecho). [ÿ]

AVACT was calculated by using the LVOT area measured on CT (LVOT area approximated by πr2 with the average diameter used for r) in the continuity equation with VTI at LVOT and transaortic flow on echocardiography (a hybrid of measures from both CT and echocardiography; AVACT = LVOTCT × VTILVOT/VTIAo). 

E-4. You defined classic and paradoxical low flow low gradient AS in your methods along with severe AS and moderate AS – please present baseline/echo parameters in table 1 based on these subgroups also. Was there an association between the LFLG AS subgroups and AVA discordance? What about calcium score (which can distort AVA-CT measurement) is it associated with the AVA discordance?

Response: Thank you for your comment. The echocardiographic parameters based on the Degree of AS (Severe AS [high-gradient severe AS, classic LF-LG AS and paradoxical LF-LG AS] and moderate AS) and related subgroups are demonstrated below. This table was included as a Supplementary table in our revised manuscript for clarity. 

Among the patients with severe AS (n = 511), the discordant group was more frequent in patients with LF-LG AS compared with those with high-gradient severe AS (P = 0.04; Table 2 and Supplementary Table 1 in revised manuscript). In Table 3, no significant difference was observed between concordant and discordant groups in terms of AVA calcium score (2747.90 [IQR, 1523.28 – 4153.45] vs. 2229.95 [1422.80 – 3734.95]; P = 0.18). The AVA calcium score which was calculated at the burden of calcification at the level of valve and the aortic cusps and does not appear to have a direct effect on the discordancy between AVACT or AVAecho. This is likely due to both measure areas being based on the continuity equation by applying the value of LVOT level on CT or echocardiography.

E-5. Add the following characteristics to table 1: diabetes, coronary heart disease, aortic aneurysm, NYHA class, echo: LVSVi, dimensionless index, LAVi, RVSP, RV function, CT: LVEDVi, LVESVi, LVSVi, LVEF; bioprosthetic or mechanical AVR, concomitant surgery, operative mortality (30-days).

Response: Thank you for your comment. We have added additional clinical information (diabetes, coronary heart disease, aortic aneurysm, concomitant surgery and valve type) in Table 1. Operative mortality (30-day mortality) was found to be 1.5% (8/535). 

We apologize that unfortunately we do not have available information for the other characteristics you mentioned above, NYHA class, RV function and LV function on echocardiography and/or CT. 

 

E-6. In the Bland-Altman plot figure, please use average AVA echo and CT in the X axis.

Response: We thought it would be more informative to demonstrate the X axis with the absolute value of echocardiography rather than the mean of two modalities. In our revised manuscript, we have replaced the new Bland-Altman plot displaying the X axis with the average for echo and CT as below. 

Revised manuscript (Figure 3): Bland-Altman plot of AVAecho and AVACT (with the X-axis showing the mean of measures from echocardiography and CT). 

E-7. Another figure illustrating the AVAecho and AVA CT technique used would be appreciated for the journal’s medical audience.

Response: We followed your comment and add a figure (Figure 2) illustrating the measurement of AVAecho and AVCCT used in this study. 

Revised manuscript (Figure 2): 

Figure 2. An example of aortic valve area (AVA) measurement using echocardiography or CT in a 74-year-old male who diagnosed as severe stenosis of fused bicuspid aortic valve.

AVA, aortic valve area; LVOT, left ventricular outflow tract; VTI, velocity time integral.

E-8. Perform multivariable analysis of clinical, echo and CT parameters to predict overall mortality during follow-up (remember up to 7 covariates given 74 mortality events). Was CT valve parameters stronger predictor than echo?

Response: Thank you for your comment. As recommended, we have performed an analysis using Cox proportional hazard models for the prognostic value of echo and CT parameters. However, both AVAecho and AVACT did not show a significant prognostic value for predicting either overall mortality or MACCE adjusted by age, sex and history of PIC or CABG (AVAecho: adjusted HR, 1.00 [95%CI, 0.99−1.01], P = .549; AVACT: adjusted HR, 1.01 [95%CI, 1.00−1.01], P = .080; AVAplani: adjusted HR, 1.00 [95%CI, 0.99−1.01], P = .561; discordant group: adjusted HR, 1.25 [95%CI, 0.68−2.32], P = .472;). These results may be due to the fact that our cohort is limited to patients who underwent AVR and therefore does not evaluate the effect on mortality due to the natural course of disease according to AS severity. In addition, patients with mild AS were excluded from the study cohort which may also limit the scope of the results. 

In a previous study carried out by Clavel et al. (Reference 13), AVAecho or AVACT were independently predictive (hazard ratio [HR]: 1.26, 95% confidence interval [CI]: 1.13 to 1.42; P < 0.0001 or HR: 1.18, 95% CI: 1.09 to 1.29 per 0.10 cm2 decrease; P < 0.0001) with a similar prognostic value in patients with isolated calcific AS. Therefore, we believe in future studies, a comprehensive evaluation would be valuable and in fact required, in order to grade the effect of AS severity using different modalities for the clinical outcome of medical management compared to surgery. This point has been added in the study limitation with further details.

Revised manuscript (Discussion): Firstly, the prognostic implications of the discordance between AVACT and AVAecho could not be addressed in this study. This resulted from the discordant results between CT and echocardiography not being considered when determining the timing of AVR. Furthermore, as echocardiography was the standard exam, the effect of discrepancies between two modalities on the patients outcome could not be evaluated. Secondly, our cohort was limited to patients who underwent AVR and therefore does not evaluate the effect on mortality due to the natural course of disease according to AS severity. As our study included patients with only a narrow higher end (severe) of AS, as opposed to the lower end (mild) of AS, this can cause selection bias. Future studies focusing on assessing the modality of choice to establish the optimal time of surgery and the effect of AS severity grading using these different modalities would be of value in order to determine the clinical outcome of medical management compared to surgery.

E-9. Have a paragraph before limitations in the discussion called clinical implications. What are the main implications for your study? When and how do you recommend CT-AVA be used and what thresholds for severe? How does your study change current management?

Response: The aim of this study was to demonstrate the factors affecting AVA discrepancies measured on echocardiography and CT. Patients with AS who underwent pre-operative echocardiography and CT for aortic valve replacement were evaluated. Of the patients analysed, ~11% of those with severe AS based on echocardiography showed discordant grading of severe AS between echocardiography and CT. Patients with larger BSA, AVAecho, tricuspid valve morphology, and annulus area normalized to the BSA were significant factors that were associated with discordant grading of AVA. The combined use of CT and echocardiography in severe AS patients should be emphasized and might be helpful for AS grading. A recent study demonstrated that surgical intervention could improve survival even among asymptomatic patients with severe AS [4]. Therefore, the decision to perform surgery in an asymptomatic patient further highlights the accurate stratification of severity of AS in those patients. 

Reference

4. Kang DH, Park SJ, Lee SA, Lee S, Kim DH, Kim HK, et al. Early Surgery or

Conservative Care for Asymptomatic Aortic Stenosis. The New England journal of

medicine. 2020;382(2):111-9.

E-10. Any items you can’t address above need to add to limitations.

Response: Thank you for your valuable comment which was made to help improve our paper. We have revised the manuscript as described above.

Comments from Reviewer #1 

1-Retrospective study.

2-Data collected from medical records

R1-1. 3-Methods section lacks details on how the AVA by CT was calculated. Did the authors used the CT maximal LVOT diameter, average area or planimetered LVOT area?

Response: Thank you for your comment. In response, we have added an additional description for the AVACT in our revised manuscript. AVACT was calculated by using the LVOT area measured on CT in the continuity equation with VTI at LVOT and transaortic flow on echocardiography (a hybrid of measures from both CT and echocardiography; AVACT = LVOTCT × VTILVOT/VTIAo). The LVOT area was determined by πr2 using a circular LVOT area , in which an average was used for the diameter.

Revised manuscript (Methods): AVACT was calculated by using the LVOT area measured on CT (LVOT area approximated by πr2 with average diameter used for r) in the continuity equation with VTI at LVOT and transaortic flow on echocardiography (a hybrid of measures from both CT and echocardiography; AVACT = LVOTCT × VTILVOT/VTIAo). 

R1-2. 4-Discordant valve area was defined when the difference between echo and CT was>0.2cm2. This occurred in a relatively small number of patients.

Response: We used a 1.2 cm2 cut-off for diagnosing severe AS on CT and a 1.0 cm2 cut-off for echocardiography in accord with the previous literature (Reference 13). These cut-off values are also widely used and accepted by most clinicians and researchers. The discordant grading for severe AS occurred in 10.96% (56/511) of patients with severe AS. This is because the two modalities (CT and echocardiography) are well correlated to measure the AVA; however, a value of ~10% as a total of severe AS patients is not small or a trivial matter and do not deserve to be neglected in clinical practice. Moreover, although the number of patients were small according to these criteria, it was meaningful to evaluate the clinical characteristics of patients with discordant severity grading between two modalities for AVA.

Reference 

13. Clavel MA, Malouf J, Messika-Zeitoun D, Araoz PA, Michelena HI, Enriquez-

Sarano M. Aortic valve area calculation in aortic stenosis by CT and Doppler

echocardiography. JACC Cardiovasc Imaging. 2015;8(3):248-57.

R1-3. 5-As in prior studies AV by CT was largen than echo AVA.

Response: Yes, our results was consistent with previous studies (reference 13), in that AVACT was greater than AVAEcho. 

Reference 

13. Clavel MA, Malouf J, Messika-Zeitoun D, Araoz PA, Michelena HI, Enriquez-

Sarano M. Aortic valve area calculation in aortic stenosis by CT and Doppler

echocardiography. JACC Cardiovasc Imaging. 2015;8(3):248-57.

R1-4. 6-The main objective of the study was to find the determinants form discrepant AVA

Response: Yes, correct. The main aim of this study was to find and demonstrate the factors affecting AVA discrepancies measured on CT and echocardiography.

R1-5. 7-In the bland Altman plot shown, the X axis shows the echo AVA and not the average of Echo and CT as is typically done.

Response: We agree that typically, the average of both measures is used in a Bland-Altman plot and we can also change the X axis to plot the average of the echo and CT data. However, as demonstrated in response to E-6, we wanted to demonstrate the absolute value of echocardiography rather than the mean of measures using two modalities to be more exact. We have now revised the Bland-Altman plot with the average of the echo and CT as the X-axis value below for Figure 3. However, the value of ICC does not appear to change. 

Revised manuscript (Figure 3): Bland-Altman plot of AVAecho and AVACT (X-axis shows a mean of measures from echocardiography and CT). 

R1-6. 8-Not clear what the unique contribution to this paper compared to similar ones that already exist in the literature.

Response: There were several previous studies that compared the AVAecho and AVACT; however, there was no demonstration that certain patient factors can affect the discrepancy between two measures. Although currently echocardiography plays the primary role in evaluating AVA prior to make a decision for AVR or TAVI, measurements using echocardiography can be inaccurate in certain cases. In such cases, CT can also provide anatomical information as a complementary tool. Moreover, when evaluating AVA in patients with low-flow/low-gradient AS, there can be discordant results based on two modalities for grouping the patients to either true severe AS or not. Thus, it is imperative to understand the differences as well as the pitfalls of the two modalities and which patient factors can result in the discordant grading for AS based on the two modalities in clinical practice.

Comments from Reviewer #2

The manuscript entitled "Differences in Aortic Valve Area Measured on Cardiac CT and Echocardiography in Patients with Aortic Stenosis" by Koo. et al. aimed to compare the aortic valve area AVA measured on CT and echocardiography, and demonstrate the factors that lead to discordant AVA readings between CT and echocardiography. From the 781 retrospectively selected patients, 95.51% were documented to have echocardiographic severe aortic stenosis and 10.96% of those had discordance between AVAecho and AVACT. Larger BSA, AVAecho, tricuspid AV morphology and indexed annulus area were associated with AVA discordance between echo and CT.

The authors have done a lot of work in collating this information and improving our knowledge in multimodality AS classification with few clarifications requested.

R2-1. It should be highlighted in the text that the main comparison between AVAecho and AVACT utilizes continuity equation for both and thus are comparisons of an echo solely or a hybrid approach. This said, a hybrid approach measurement can also be simplified to the comparison of annular area in both modalities.

Response: Thank you for your comment. We also agree and we tried to clarify those points in the revised manuscript, both the Abstract and Methods section. AVA on echocardiography was calculated using a continuity equation with VTI. We added an additional description for the AVAecho and AVACT in our revised manuscript. AVACT was also calculated by using a hybrid method incorporating information from both echocardiography and CT, with the LVOT area measured on CT in the continuity equation with VTI at LVOT and transaortic flow on echocardiography (AVACT = LVOTCT × VTILVOT/VTIAo). For the LVOT area, a circular LVOT area was estimated by πr2 using an average diameter. 

Revised manuscript (Abstract): AVA was obtained by AVA on echocardiography (AVAecho) and CT (AVACT) using a measurement of the left ventricular outflow tract on each modalities and correlations between those measures were evaluated.

Revised manuscript (Methods): 

The maximal and mean pressure gradients (PG) across the AV were estimated using a modified Bernoulli equation, and the AVA was calculated from the continuity equation with VTI (AVAecho). [ÿ]

AVACT was calculated by using the LVOT area measured on CT (LVOT area approximated by πr2 with the average diameter used for r) in the continuity equation with VTI at LVOT and transaortic flow on echocardiography (a hybrid of measures from both CT and echocardiography; AVACT = LVOTCT × VTILVOT/VTIAo). 

R2-2. Furthermore, the discussion section sheds less light on AVAplani as a comparison and thus is confusing as the norm of a CT approach is usually to check planimetry area measurements. If that is not the case, it is then important to emphasize the differences within CT measurements whether measurement have been taken by the hybrid approach or by planimetry.

Response: Thank you for your valuable comment. Accurately measuring AVA using a planimetry approach is limited due to the difficulty in drawing the maximal systolic opening area on 2D-thin slice thickness images as the tips of the aortic cusps are not contained to one 2D plane. On the other hand, using thick slice thickness images for planimetry area measurements makes it challenging to outline the margin of cusps due to the blurring of images which can lead to overestimating the valve opening area. Therefore, calculating AVA by incorporating velocity information from echocardiography and the size of LVOT on high spatial resolution cardiac CT images based on continuity equations can be more accurate to reflect the maximal systolic opening of AV. We further discuss this point in the Discussion section as shown below.

Revised manuscript (Discussion): In this study, the correlation between AVAecho and AVACT showed a high positive correlation, whereas the correlation between AVAplani and AVAecho was moderate (r, 0.79 vs. 0.52). As a result of this finding, we further defined discordant and concordant groups based on AVACT by the continuity equation rather than AVAplani. Accurately measuring AVA using a planimetry approach is limited due to the difficulty in drawing the maximal systolic opening area on 2D-thin slice thickness images as the tips of the aortic cusps are not contained to one 2D plane. On the other hand, using thick slice thickness images for planimetry area measurements makes it challenging to outline the margin of cusps due to the blurring of images which can lead to overestimating the valve opening area. Therefore, calculating AVA by incorporating velocity information from echocardiography and the size of LVOT on high spatial resolution cardiac CT images based on continuity equations can be more accurate to reflect the maximal systolic opening of AV.

- End -

---

## [Editor Report · Decision Letter 1]

2 Jan 2023

Differences in Aortic Valve Area Measured on Cardiac CT and Echocardiography in Patients with Aortic Stenosis

PONE-D-22-24629R1

Dear Dr. Koo,

We’re pleased to inform you that your manuscript has been judged scientifically suitable for publication and will be formally accepted for publication once it meets all outstanding technical requirements.

Kind regards,

Tom Wang

Academic Editor

PLOS ONE

Additional Editor Comments (optional):

Thank you for adequately addressing the reviewers' and editor's comments.
---

## [Editor Report · Acceptance letter]

10 Jan 2023

PONE-D-22-24629R1 

Differences in Aortic Valve Area Measured on Cardiac CT and Echocardiography in Patients with Aortic Stenosis 

Dear Dr. Koo:

I'm pleased to inform you that your manuscript has been deemed suitable for publication in PLOS ONE. Congratulations! Your manuscript is now with our production department. 

Kind regards, 

on behalf of

Dr. Tom Kai Ming Wang 

Academic Editor

PLOS ONE